# Artificial Intelligence in the Management of Barrett’s Esophagus and Early Esophageal Adenocarcinoma

**DOI:** 10.3390/cancers14081918

**Published:** 2022-04-10

**Authors:** Franz Ludwig Dumoulin, Fabian Dario Rodriguez-Monaco, Alanna Ebigbo, Ingo Steinbrück

**Affiliations:** 1Department of Medicine and Gastroenterology, Gemeinschaftskrankenhaus Bonn, Academic Teaching Hospital, University of Bonn, Bonner Talweg 4-6, D-53113 Bonn, Germany; f.rodriguez@gk-bonn.de; 2Department of Gastroenterology, University Hospital Augsburg, Stenglinstr. 2, D-86156 Augsburg, Germany; alanna.ebigbo@uk-augsburg.de; 3Department of Medicine, Evangelisches Diakoniekrankenhaus Freiburg, Academic Teaching Hospital, University of Freiburg, Wirthstraße 11, D-79110 Freiburg, Germany; ingo.steinbrueck@diak-fr.de

**Keywords:** artificial intelligence, gastro-esophageal reflux disease, Barrett’s esophagus, early adenocarcinoma of the esophagus, deep convolutional neuronal networks

## Abstract

**Simple Summary:**

Esophageal adenocarcinoma is increasing in incidence and is the most common subtype of esophageal cancer in Western societies. AI systems are currently under development and validation in many fields of gas-troenterology.

**Abstract:**

Esophageal adenocarcinoma is increasing in incidence and is the most common subtype of esophageal cancer in Western societies. The stepwise progression of Barrett´s metaplasia to high-grade dysplasia and invasive adenocarcinoma provides an opportunity for screening and surveillance. There are important unresolved issues, which include (i) refining the definition of the screening population in order to avoid unnecessary invasive diagnostics, (ii) a more precise prediction of the (very heterogeneous) individual progression risk from metaplasia to invasive cancer in order to better tailor surveillance recommendations, (iii) improvement of the quality of endoscopy in order to reduce the high miss rate for early neoplastic lesions, and (iv) support for the diagnosis of tumor infiltration depth in order to guide treatment decisions. Artificial intelligence (AI) systems might be useful as a support to better solve the above-mentioned issues.

## 1. Introduction

Esophageal adenocarcinoma is among the cancer entities with increasing incidence and is the most common subtype of esophageal cancer in Western societies [1]. The majority of patients are still diagnosed at advanced stages, with 5-year survival rates below 20% [2]. Barrett’s metaplasia, defined as the replacement of squamous epithelium of the distal esophagus by metaplastic (intestinal-type) epithelium due to gastro-esophageal reflux, is a potential precursor for esophageal adenocarcinoma. It is associated with chronic reflux, with an estimated prevalence of 1–2% in the general population and up to 10% in patients with chronic reflux disease. The stepwise progression of Barrett’s metaplasia to high-grade dysplasia and invasive cancer provides an opportunity for screening and surveillance. In fact, the rapid development of endoscopic imaging and minimally invasive treatment could potentially decrease mortality from Barrett’s-associated adenocarcinoma [3,4,5]. Unfortunately, there are many unresolved issues, resulting in different recommendations from different guidelines [6]. The prevalence of Barrett’s esophagus in the general population is low and symptoms of reflux disease are an unreliable marker for metaplasia, which makes the decision for invasive (endoscopic) screening difficult. Moreover, the risk of progression from non-dysplastic Barrett’s esophagus to adenocarcinoma is very heterogeneous. Finally, the accuracy of upper gastrointestinal (GI) endoscopy is not optimal given the high miss rate for neoplastic lesions. 

Artificial intelligence (AI) is broadly defined as machine learning and, in its most advanced form, uses a convolutional neuronal network (CNN) structure. Among other possible applications, AI systems can be used for object identification. They can be trained to identify the presence of an abnormal structure (e.g., to recognize an image pattern suggestive of neoplasia) and to perform the segmentation task (i.e., localize the abnormal structure). Most of the previous work has been done on the detection of premalignant and malignant lesions [7]. Thus, AI systems increase the diagnostic accuracy of upper and lower gastrointestinal endoscopy. AI has also been useful in the prediction of gastrointestinal cancer prognosis, or to identify patient characteristics associated with treatment responses. Other applications include the detection of possible bleeding sources in small bowel capsule endoscopy or the differentiation of dysplastic from inflammatory lesions in chronic inflammatory bowel disease [7]. At the time of writing, AI systems for colorectal polyp detection are already commercially available in several countries. Nevertheless, the application of AI systems for Barrett’s esophagus (i.e., not only the detection and characterization of neoplastic or dysplastic lesions) is less advanced. This review will therefore focus on AI systems that might become useful tools to help with the above-mentioned issues in the management of Barrett’s esophagus.

## 2. Optimizing Screening for Barrett’s Metaplasia 

### 2.1. AI Systems to Ameliorate the Quality of Upper Gastrointestinal (GI) Endoscopy

While there has been an increase in the incidence of gastro-esophageal reflux disease (GERD), Barrett’s esophagus, and esophageal cancer, the current guidelines from the American Society for Gastroenterology (ASGE) state that there is insufficient evidence to support general screening for Barrett’s esophagus [1]. Nevertheless, a larger number of endoscopies of the upper gastrointestinal tract are carried out for other indications, providing an opportunity to incidentally detect neoplastic lesions. Unfortunately, the quality of these procedures is far from perfect. Data from a retrospective analysis of more than 120,000 upper GI endoscopies performed in four tertiary centers for various indications (excluding high-risk populations) revealed a 6.4% miss rate for esophageal cancer, with an associated two-year survival rate of only 20% [8]. In an attempt to support the quality assessment of upper GI endoscopies, an AI system was established with over 96% accuracy for the detection and classification of upper gastrointestinal anatomy. The system was then used to evaluate the completeness of photo documentation from 472 upper endoscopies. Surprisingly, the complete documentation rate from the esophagus to the duodenum was as low as 78.0%; endoscopists with a higher colorectal adenoma detection rate had a better performance [9]. In another study, a fully convolutional neural network was developed to automatically identify Barrett’s metaplasia in endoscopic still images [10]. The performance of the system for segmentation was expressed as ‘intersection over union’ (IOU (or Jaccard Index) quantifies the concordance between a predicted and true localization of an object under study. It is calculated as the overlap of predicted segmentation and ground truth divided by the sum of both images). The system detected Barrett’s metaplasia in endoscopic images with IOU values of 0.56 for the gastro-esophageal junction and 0.82 for the squamo-columnar junction, respectively. After further refinement, both systems could be useful tools to enhance the diagnostic performance of upper GI endoscopy. 

### 2.2. Identification of Individuals at Risk for Barrett’s Esophagus for Invasive (Endoscopic) Screening

The current guidelines from the ASGE state that if endoscopic screening is contemplated it should be performed in an at-risk population, for example, in individuals with a family history of esophageal adenocarcinoma or chronic reflux disease with an associated risk factor [1]. A recent systematic review and meta-analysis of more than 300,000 endoscopies with various indications supported these risk factors: the prevalence of Barrett’s metaplasia ranged from 0.8% (general population) and 3% (individuals with reflux symptoms) up to 12% (patients with reflux disease and additional risk factors) and even 23% (positive family history of esophageal adenocarcinoma) [11]. Unfortunately, a systematic formal evaluation of the screening recommendations for Barrett’s esophagus from different gastroenterological societies showed low sensitivities (38.6–43.2%) if reflux symptoms were used as a trigger. On the other hand, guidelines also relying on other risk factors had low specificities and a low area under the receiver operating characteristic curve (AUROC) of 0.50–0.60 [12]. To overcome these limitations, the ‘MARK-BE’ study used different machine learning methods to create a stable algorithm in order to better predict the presence of Barrett’s metaplasia or adenocarcinoma from patient characteristics. Different machine learning classification algorithms were fed with well-characterized datasets from two large case–control studies. A panel of 8/40 questionnaire items showed independent diagnostic relevance (age, gender, smoking, waist circumference, frequency of stomach pain, duration of heartburn, acid taste, acid suppression therapy). An algorithm using logistic regression had the highest performance for the prediction of Barrett’s metaplasia, with an AUROC of 0.87. Inclusion of more than eight features in the algorithm did not result in a more precise prediction [13].

In addition, to optimize the yield of screening endoscopy using patient characteristics to predict Barrett’s metaplasia or adenocarcinoma, there has been some interest in less-invasive screening procedures. A proof-of-principle study used an artificial nose device to characterize the pattern of exhaled volatile compounds for the prediction of Barrett’s esophagus. The device consisted of an AI-trained artificial nose and was evaluated against a background of normal and GERD patients in 402 cases. The test could distinguish between individuals with and without Barrett’s esophagus with good diagnostic accuracy (sensitivity 91%, specificity 74%). After further refinement, this test might become an efficient screening method to identify high-risk individuals for subsequent diagnostic upper endoscopy [14]. Another approach to non-endoscopic screening is the cytosponge trefoil factor-3 (TFF-3) procedure, which analyzes a cytology specimen from the upper gastrointestinal tract [15]. To obtain the specimen, a gelatin capsule containing a compressed cytosponge is swallowed. After the sponge is released and has expanded in the stomach, it is extracted with an attached string. The resulting brush cytology is then analyzed for TFF-3, which is a marker for the goblet cells characteristic of Barrett´s esophagus. The cytological analysis can be supported by a semi-automated AI system, which reduces the pathologist’s workload without compromising overall test performance [16]. 

### 2.3. Optimizing Surveillance in Patients with Barrett’s Esophagus

All current guidelines recommend endoscopic surveillance once a diagnosis of Barrett’s esophagus has been established [1,3,6]. The goal of this strategy is to identify early neoplastic changes suitable for endoscopic resection (Figure 1). The risk of progression for Barrett’s esophagus varies considerably, and a number of demographic risk factors (e.g., age, gender, race, family history of esophageal adenocarcinoma, longstanding reflux symptoms, smoking) as well as characteristics of the metaplasia itself (e.g., length of the segment, presence and grade of dysplasia) have been identified and integrated into a predictive score [17]. All guidelines recommend the Prague (CM) classification to semi-quantitatively assess the extent of metaplasia at endoscopic surveillance [6]. However, it is sometimes difficult to accurately determine the extent of Barrett’s metaplasia in real life, particularly for non-expert endoscopists. Therefore, an AI system has been used to automatically extract these measurements during endoscopy. The performance of the system was very good, with an accuracy greater than 97% for CM measurements on 3D phantom images. This AI system could provide a more reliable classification and—perhaps more importantly—a more reliable image set for follow up [18].

#### 2.3.1. Improving the Detection of Neoplasia on High-Definition White Light Endoscopy (HD-WLE) 

The ultimate goal of endoscopic surveillance is the detection of early neoplastic chances suitable for minimally invasive endoscopic treatment. Guidelines recommend surveillance with high-definition white light endoscopy, virtual and dye-assisted chromoendoscopy, and targeted biopsy as well as Seattle protocol biopsies (four-quadrant biopsies every 1–2 cm of Barrett’s segment). Unfortunately, these screening procedures are not very effective. A systematic review and meta-analysis published in 2016 showed that 25% of esophageal adenocarcinomas were diagnosed within a year after an upper GI endoscopy diagnosing Barrett’s metaplasia [19]. Moreover, a recent meta-analysis of seven studies validating AI systems also showed a relatively low sensitivity and specificity for the detection of upper GI tract cancer [20]. Therefore, several recent studies have investigated the potential of AI systems to improve the detection rate of early neoplasia on HD-WLE images and real-time videos, and on advanced imaging modalities such as volumetric laser endomicroscopy and spectral endoscopy.

The most straightforward approach would be a computer-assisted diagnosis (CAD) system, supporting the detection of neoplastic lesions during real-time endoscopy. Such a system, if well-trained by expert endoscopists, could increase the performance of any endoscopist by marking the area of interest for further optical evaluation, biopsy, or both. Ideally, the system should not only reliably allow for classification (binary discrimination: non-dysplastic vs. dysplastic) and segmentation (delineation of the dysplastic area), but should also be able to perform these tasks in real-time (with a frame rate comparable to that of standard video endoscopy—i.e., ca. 70 frames/second). Eight studies have evaluated the performance of such systems on still images and, in some pilot studies, on real-time video endoscopy. All groups used deep learning methods to train their AI systems with high-quality images annotated by experts. Pioneering work from Augsburg analyzed the performance of a deep learning system to detect and delineate early adenocarcinoma (Figure 2). The performance for HD-WLE on two different image sets yielded sensitivities and specificities of 92–97% and 88–100%, respectively. One image set was also evaluated for narrow-band imaging (NBI), with a sensitivity and specificity of 94% and 80%, respectively. Using tumor margins delineated by experts, the Dice coefficient (F1 score) was 0.72, indicating a good overlap between predicted segmentation and ground truth [21]. In a subsequent study, this AI system was evaluated during video endoscopies of 14 cases with neoplastic Barrett’s esophagus and showed an impressive predictive accuracy of 89.9% [22]. In a similar study, a deep learning system was developed, validated, and subsequently compared with the performance of a group of international experts. The system detected Barrett’s neoplasia in still images with a higher accuracy than experts and showed a near-perfect delineation of the lesions [23]. The system was also tested in a pilot study during real-time endoscopy on 20 patients with a comparable performance (accuracy, sensitivity, and specificity of 90%, 91%, and 89%, respectively) [24]. In another study, a trained CNN detected early Barrett’s neoplasia with high sensitivity, specificity, and accuracy in real-time videos. The system could also locate the dysplastic areas with high precision and speed [25]. Finally, a Japanese group developed an artificial intelligence system for the diagnosis of early cancer at the esophageal-gastric junction. A total of 1172 images from 166 cancer cases and 2271 images of normal mucosa from 219 patients were used as training data. The system was then validated using 232 images from 36 cancer cases and 43 non-cancerous lesions. Sensitivity, specificity, and accuracy of the AI system (94%, 42%, and 66%, respectively) compared favorably to that of the performance of board-certified experts [26] (Table 1). The performance of different popular object detection algorithms (all using CNN) was compared using 100 HD-WLE images from 39 patients annotated by experts as ground truth. The single-shot multibox detector (SSD) algorithm outperformed other methods, with a sensitivity of 96% and a specificity of 92% for the identification of early esophageal adenocarcinoma [27].

#### 2.3.2. Volumetric Laser Endomicroscopy (VLE) and Spectral Endoscopy

VLE is a balloon-based system that allows for the real-time optical coherence tomography of large (6 cm) segments of esophagus. Suspicious areas can be marked for later tissue sampling [28]. Data from a US registry with 1000 patients showed that the addition of VLE to the standard of care improved neoplasia detection by 55% [29]. CAD systems are currently under development, using AI to optimize the analysis of VLE images [30,31,32]. The largest dataset published thus far is from a multicenter study that established and evaluated an AI algorithm for the VLE-based detection of Barrett’s neoplasia. During VLE, 229 non-dysplastic and 89 neoplastic lesions from 47 patients were marked for subsequent biopsy. The first 22 patients (134 non-dysplastic/38 neoplastic lesions) were used as a training set, the remainder were used for validation and comparison with the diagnostic performance of experts. Accuracy, sensitivity, and specificity for the validation were 85%, 91%, and 82%, respectively. Moreover, the AI system outperformed the VLE experts [33].

Waterhouse et al. evaluated the performance of a custom-built spectral (baby) endoscope, which is introduced through the working channel of a therapeutic scope. This endoscope allows a more comprehensive analysis of spectral information beyond conventional HD-WLE. This prototype, which has high accuracy with up to 12-fold contrast improvement relative to WLE, could be used for the discrimination of neoplastic versus non-neoplastic lesions. In particular, the system was able to classify non-dysplastic versus dysplastic areas with 84.8% accuracy [34].

#### 2.3.3. The Wide-Area Transepithelial Sampling Three Dimensional (WATS 3D) Procedure

The WATS procedure combines specifically designed brush cytology with semi-automated 3D cytopathology analysis [1,28]. The WATS-3D procedure is recommended in the current ASGE guidelines [1]. For cytopathology, a computer-driven algorithm highlights suspicious areas to better identify dysplasia. This procedure increases the detection rate of dysplasia in Barrett’s metaplasia [35,36]. 

#### 2.3.4. AI to Determine the Infiltration Depth of Neoplastic Lesions

Finally, AI could also be helpful in decision-making during the visual evaluation of a malignant lesion. Infiltration depth—along with other factors such as grading, vessel infiltration, and resection status—is an important predictor of lymph node metastasis. Current visual evaluation of infiltration depth is not very precise. Therefore, a recent study evaluated 230 white-light endoscopic images of Barrett’s-associated adenocarcinoma (108 pT1a and 122 pT1b) from expert centers. Using the above-mentioned deep learning system, a fair accuracy for the discrimination of pT1a versus pT1b Barrett’s-associated adenocarcinoma was demonstrated [37]. Thus, AI systems could also support decision-making regarding the endoscopic versus surgical resection of early cancerous lesions. 

## 3. Conclusions

AI systems are currently under development and validation in many fields of gastroenterology. They are likely to become an important tool to support the management of Barrett’s esophagus. In particular, they might be helpful for the management of different aspects of the condition, e.g., improving the quality assessment of upper GI endoscopy, identifying individuals for endoscopic screening, predicting the risk for histologic progression, enhancing the diagnostic yield of surveillance endoscopies, and even predicting infiltration depth when evaluating a cancerous lesion for possible endoscopic treatment. Regarding the latter issues, AI-enhanced systems to detect neoplastic chances in Barrett’s metaplasia will soon be ready to assist routine endoscopy. Further development and spread of these AI systems will hopefully result in more consistent and earlier detection of neoplastic changes. Finally, in combination with the increasing possibilities of organ-preserving treatment options for early neoplasia, the technology could lead to a decrease in morbidity and mortality from Barrett’s-associated adenocarcinoma.

## Figures and Tables

**Figure 1 cancers-14-01918-f001:**
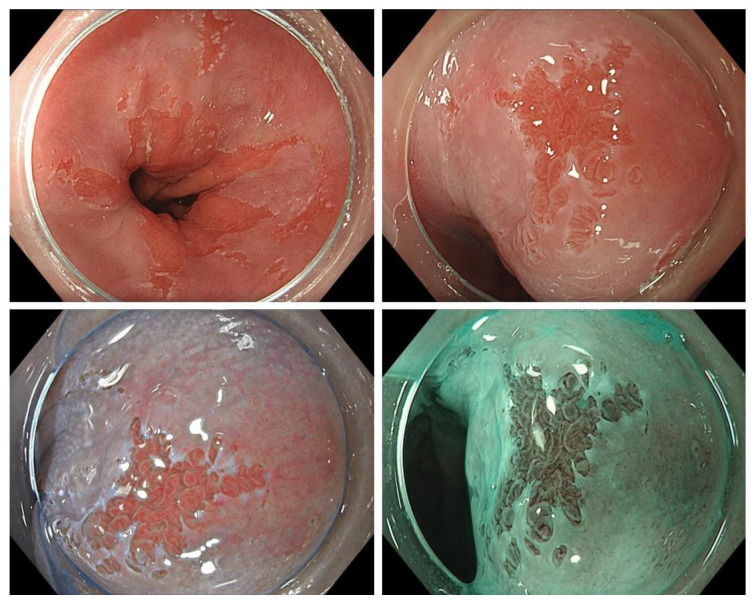
Example of a small Barrett’s adenocarcinoma detected during surveillance endoscopy (**top left**: overview of the transition zone; **top right**: close-up view (HD-WLE); **bottom left**: close-up view (texture and color enhancement imaging/TXI); **bottom right**: close-up view, narrow-band imaging (NBI) mode). Images from the Olympus Evis X1 endoscopy system.

**Figure 2 cancers-14-01918-f002:**
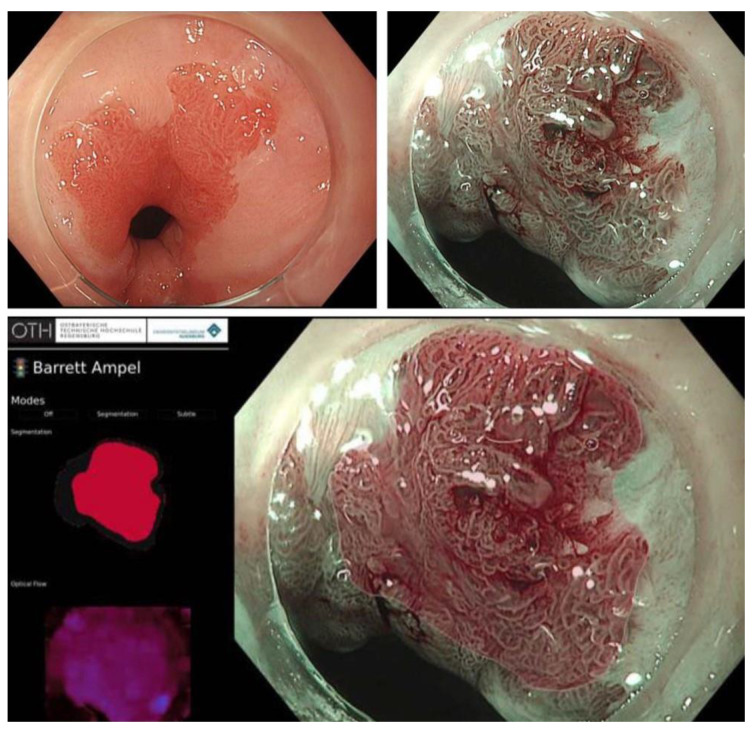
Example of a small Barrett’s-associated adenocarcinoma detected by the Augsburg artificial intelligence system [21]. **Top right**: close-up view (HD-WLE). **Top left**: close-up view (NBI mode). **Bottom**: functionality of the ‘Barrett traffic light’, which gives a heat map indicating areas suspicious of dysplasia in red—corresponding to the endoscopic image on the right side of the picture.

**Table 1 cancers-14-01918-t001:** Overview of current studies on AI-supported detection of Barrett’s neoplasia.

** Study **	** Main Findings **
**Ebigbo et al.,** **2019** **[21]**	Objective Detection of Barrett’s-associated adenocarcinoma (still images) Datasets (HD-WLE and NBI images) MICCAI data (100 images: 17 neoplastic, 22 non-neoplastic) Augsburg data (148 images: 33 neoplastic, 41 non-neoplastic) Performance (binary task: detection of neoplasia) WLI: sensitivity 92 and 97%/specificity 88 and 100% NBI: sensitivity 94%/specificity 80% AI system outperformed 13 experts Performance (object identification: localization of neoplasia) Dice coefficients ^1^ of 0.72 and 0.56 for the two datasets
**Ebigbo et al.,** **2020** **[22]**	Objective Detection of Barrett’s-associated adenocarcinoma (videos, no image freeze) Datasets (HD-WLE videos) Training data: 129 images Validation data: 62 images (36 neoplastic) from 14 patients Performance (binary task: detection of neoplasia) Sensitivity 83.7%/specificity 100.0%/accuracy 89.9%
**De Groof et al.,** **2020** **[23]**	Objective Detection of Barrett’s-associated adenocarcinoma (still images) Datasets (HD-WLE images) Total of 1704 images from 669 patients Training: 1544 images in two datasets (819× neoplasia) Validation: 160 images in two datasets (80× neoplasia) Performance (binary task: detection of neoplasia) Sensitivity 90%/specificity 88%/accuracy 89% AI system outperformed 53 non-expert endoscopists Performance (object identification: localization of neoplasia) AI system identified the optimal site for biopsy in 97%/92%
**De Groof et al.,** **2020** **[24]**	Objective Detection of Barrett’s-associated adenocarcinoma (videos, image freeze) Datasets (HD-WLE videos) Validation: 20 videos (10× neoplasia) Performance (binary task: detection of neoplasia) Sensitivity 91%, specificity 89%, accuracy 90%
**Hashimoto et al.,** **2020** **[25]**	Objective Detection of Barrett’s-associated adenocarcinoma (real-time videos) Datasets (HD-WLE videos) Total of 2 × 916 images from 100 patients (70× neoplasia) Training: 1374 (691× neoplasia) Validation: 458 (225× neoplasia) Performance (binary task: detection of neoplasia) Sensitivity 96.4%, specificity 94.2%, accuracy 95.4% Performance (object identification: localization of neoplasia) Mean average precision 0.75/intersection over union (IOU) 0.3
**Iwagami et al.,** **2021** **[26]**	Objective Detection of cancer at the esophago-gastric junction (still images) Datasets (HD-WLE images) Total of 2 × 916 images from 100 patients (70× neoplasia) Training: 3443 images (1172 neoplasia)/385 patients (166 neoplasia) Validation: 232 images (36 cancer patients/43 controls) Performance (binary task: detection of neoplasia) Sensitivity 94%, specificity 42%, accuracy 66% The AI system outperformed 15 experts

^1^ The Dice coefficient (or F1 Score) is a measure of concordance between predicted and true localization of an object under study (e.g., minute cancer in Barrett’s esophagus). It is calculated as 2 × overlap of predicted segmentation + ground truth divided by the sum of pixels in both images.

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
