# Peer review of "Artificial Intelligence in the Management of Barrett’s Esophagus and Early Esophageal Adenocarcinoma"

_cancers, 2022, doi:10.3390/cancers14081918_

Round 1
Reviewer 1 Report
Dear Authors,
I would like to congratulate you on the submitted manuscript. The analysis on existent and emerging AI systems in detection of Barret’s Esophagus and Esophageal Adenocarcinoma is highly relevant in the actual context, and your review gives a good view of the topic.
While reading your manuscript, I have, however, observed minor aspects that could improve its quality even further. Please consider revising the following aspects:
Abstract:
Line 14: only a single “+” should appear in the phone number.
Line 20: “…to invasive cancer, to better…” addition of a comma is necessary for improving readability.
Line 22: “…lesions, and (iv)…” Oxford comma could improve readability.
- Introduction:
Line 40: “Barrett ́s cancer” is not usually observed as terminology to my knowledge. Please use a more common way to address the pathology, such as “Barrett's esophagus progression into esophageal adenocarcinoma” or alternatives.
Line 42: “…the prevalence of Barrett ́s in the general population is low …” please keep on referring to “Barrett’s esophagus” without shortening the name of the pathology in colloquial manner.
Line 43-44: “…which makes the decision for invasive…” readability would be improved by articulating “the decision”
Line 45: Please keep on referring to “Barrett’s esophagus” without shortening the name of the pathology in colloquial manner.
Line 45-46: “Finally, the quality 45 of upper GI endoscopy” in this context, “accuracy” could be a better term than “quality”. Please revise.
Line 48: “…applications, AI systems…” addition of a comma is necessary for improving readability.
Line 48: “…AI systems can be used for object identification. It can be trained…” as the 2nd sentence refers to AI systemS, please consider continuing in the 3rd person, plural: “They can be trained…”.
Line 49: “also perform the segmentation task” readability would be improved by articulating “the segmentation task”.
Based on the minor, but numerous, mistakes observed in the Abstract and Introduction sections, I highly encourage a thorough proofreading of the text in what regards English Language Editing.
- Optimizing screening for Barrett ́s metaplasia
Line 60: “…Barrett ́s esophagus, and esophageal cancer, the current ASGE guidelines state that…” Please consider the insertion and usage of the Oxford comma and the comma between sentences in the given excerpt. Moreover, “ASGE guidelines” could be more appropriate as a form in this context.
Line 109: Please spell out the term before using its abbreviation for the first time: GERD
Line 119: “reduced” should be replaced by “reduces”
Line 123: Please format references appropriately.
Figure 1. Please make sure you have all rights to use the images in the review. This can be either consent of the authors of the work from where they were collected, or the informed consent and IRB approval if they are your own.
Line 172: Please spell out the term before using its abbreviation for the first time: NBI
Line 190: “[25] (Table 1).” Please take care with formatting
Please in your work express the same metrics (i.e. sensitivity, specificity, and accuracy) in the same format. Choose either percentages or fractions, but do not switch in between at the expense of reduced readability and understanding.
Figure 2. Please obtain the consent of the authors to reuse images from their data base/publications in your work.
Line 205: “6 cm” please use common formatting with space between number and unit.
Line 218: “HD WLE” Please keep your abbreviations consistent throughout the manuscript. Previously, “HD-WLE” was used.
Line 223: Please spell out the term before using its abbreviation for the first time: WATS.
All in all, I would have liked the review to present a Conclusion/Discussion section, to bring all the findings together instead of just the Summary. In case it is possible, please consider the addition of this section.
I hope this revision finds you well and that the suggestions are constructive for improving your manuscript.
Best regards,
Author Response
Thank you very much for the thorough review of the manuscript.
- We have incorporated the suggested modification - in particular we have performed a thorough proofreading of the manuscript, have added commas, space formats, spelled out terms before using abbreviations, consistently report percentages and changed formattings in the tables (cf. highlighted changes).
- We have changed the summary to a more comprehensive conclusion section.
- The figures have not been published elsewhere. They are from Augsburg University. Patients have signed informed consent and there is an IRB (from coauthor AE).
Again we thank the reviewer for the comments and hope that we increased the quality of the manuscript.
Reviewer 2 Report
In this article, Dr Dumoulin and colleagues performed an overview of “Artificial intelligence in the management of Barrett’s esophagus and early esophageal adenocarcinoma”. The review is well-written and very interesting, I have only few remarks.
- The review is very concise (36 references). Hence, I think it would be interesting to discuss the use of Artificial Intelligence (AI) in other gastro-intestinal diseases (colon cancer, ulcerative colitis…). There are indeed some data available on the impact of AI on the miss rate of colorectal cancer, on the pathological diagnosis, or on the prognosis of colorectal cancer (notably prediction of lymph node metastasis). It is probably easier to understand what may be possible in esophageal metaplasia and dysplasia considering what has been reported in other diseases. The aim would be to include some relevant references, not to thoroughly discuss AI in the management of GI tract diseases.
- In line with my previous comment, could the authors discuss how current tools may be improved in the future or whether alternative tools may be used? This question may be naive since I am to familiar with the field of AI. Still, I think it would be important not to focus only on what has been done so far but also to discuss options and potential options.
- Please write abbreviations in full when used (ASGE L60)
Author Response
Many thanks for the helpful comments!
We have
- included a short discussion of the impact of AI systems in gastroenterology in the introduction (page 2 / 2nd paragraph)
- have discussed how current tools may impact the future of gastroenterology (conclusions page 10).
We hope the modifications will improve the quality of the manuscript.
Reviewer 3 Report
The article is a review paper on the use of AI in the management of Barrett's esophagus.
- The subject is of high clinical importance.
-The article is well written, compact.
- Statistics: not relevant.
Major recommendation;
In the paragraph 2.3.1 and line 165 is stated "eight studies have evaluated the performance of such systems-....."
However,
The first and landmark study published on AI in Barrett's is missing, and should be mentioned in the review:
Fons van der Sommen, Svitlana Zinger, Wouter L Curvers, Raf Bisschops, Oliver Pech, Bas L A M Weusten, Jacques J G H M Bergman, Peter H N de With, Erik J Schoon: Computer-aided detection of early neoplastic lesions in Barrett's esophagus. Endoscopy 04/2016; DOI:10.1055/s-0042-105284
This is of importance because the MICCAI data, which subsequent studies used and referred to, are the data from that particular study. This should be added to this review.
Author Response
Many thanks for the helpful comments. We have now included the study (top of page 6) and included the reference (21). We also mention, that the image set from that study was used in later studies.
We hope the modifications are satisfactory to the reviewer.
Kindest regards,
F.L.Dumoulin on behalf of the authors
Round 2
Reviewer 2 Report
The modifications improved the quality of the review. I have no further questions.